# LEARNING AGGREGATION FUNCTIONS

## ABSTRACT

Learning on sets is increasingly gaining attention in the machine learning community, due to its widespread applicability. Typically, representations over sets are computed by using fixed aggregation functions such as sum or maximum. However, recent results showed that universal function representation by sum- (or max-) decomposition requires either highly discontinuous (and thus poorly learnable) mappings, or a latent dimension equal to the maximum number of elements in the set. To mitigate this problem, we introduce LAF (Learning Aggregation Functions), a learnable aggregator for sets of arbitrary cardinality. LAF can approximate several extensively used aggregators (such as average, sum, maximum) as well as more complex functions (e.g. variance and skewness). We report experiments on semi-synthetic and real data showing that LAF outperforms state-of-the-art sum- (max-) decomposition architectures such as DeepSets and library-based architectures like Principal Neighborhood Aggregation.

## 1 INTRODUCTION

The need to aggregate representations is ubiquitous in deep learning. Some recent examples include max-over-time pooling used in convolutional networks for sequence classification (Kim, 2014), average pooling of neighbors in graph convolutional networks (Kipf & Welling, 2017), max-pooling in Deep Sets (Zaheer et al., 2017), in (generalized) multi-instance learning (Tibo et al., 2017) and in GraphSAGE (Hamilton et al., 2017). In all the above cases (with the exception of LSTM-pooling in GraphSAGE) the aggregation function is predefined, i.e., not tunable, which may be in general a disadvantage (Ilse et al., 2018). Sum-based aggregation has been advocated based on theoretical findings showing the permutation invariant functions can be sum-decomposed (Zaheer et al., 2017; Xu et al., 2019). However, recent results (Wagstaff et al., 2019) showed that this universal function representation guarantee requires either highly discontinuous (and thus poorly learnable) mappings, or a latent dimension equal to the maximum number of elements in the set. This suggests that learning set functions that are accurate on sets of large cardinality is difficult.

Inspired by previous work on learning uninorms (Melnikov & Hüllermeier, 2016), we propose a new parametric family of aggregation functions that we call LAF, for *learning aggregation functions*. A single LAF unit can approximate standard aggregators like sum, max or mean as well as model intermediate behaviours (possibly different in different areas of the space). In addition, LAF layers with multiple aggregation units can approximate higher order moments of distributions like variance, skewness or kurtosis. In contrast, other authors (Corso et al., 2020) suggest to employ a predefined library of elementary aggregators to be combined. Since LAF can represent sums, it can be seen as a smooth version of the class of functions that are shown in Zaheer et al. (2017) to enjoy universality results in representing set functions. The hope is that being smoother, LAF is more easily learnable. Our empirical findings show that this can be actually the case, especially when asking the model to generalize over large sets.

In particular, in this paper we offer an extensive experimental analysis showing that:

- LAF layers can learn a wide range of aggregators (including higher-order moments) on sets of scalars without background knowledge on the nature of the aggregation task
- LAF layers on the top of traditional layers can learn the same wide range of aggregators on sets of high dimensional vectors (MNIST images)
- LAF outperforms state-of-the-art set learning methods such as DeepSets and PNA on real-world problems involving point clouds and text concept set retrieval.

| Name | Definition | $a$ | $b$ | $c$ | $d$ | $e$ | $f$ | $g$ | $h$ | $\alpha$ | $\beta$ | $\gamma$ | $\delta$ | limits |
|---|---|---|---|---|---|---|---|---|---|---|---|---|---|---|
| constant | $c \in \mathbb{R}$ | 0 | 1 | - | - | 0 | 1 | - | - | $c$ | 0 | 1 | 0 | |
| max | $\max_i x_i$ | $1/r$ | $r$ | - | - | 0 | 1 | - | - | 1 | 0 | 1 | 0 | $r \to \infty$ |
| min | $\min_i x_i$ | 0 | 1 | $1/r$ | $r$ | 0 | 1 | - | - | 1 | -1 | 1 | 0 | $r \to \infty$ |
| sum | $\sum_i x_i$ | 1 | 1 | - | - | 0 | 1 | - | - | 1 | 0 | 1 | 0 | |
| nonzero count | $\lvert\{i : x_i \neq 0\}\rvert$ | 1 | 0 | - | - | 0 | 1 | - | - | 1 | 0 | 1 | 0 | |
| mean | $1/N \sum_i x_i$ | 1 | 1 | - | - | 1 | 0 | - | - | 1 | 0 | 1 | 0 | |
| $k$th moment | $1/N \sum_i x_i^k$ | 1 | $k$ | - | - | 1 | 0 | - | - | 1 | 0 | 1 | 0 | |
| $l$th power of $k$th moment | $(1/N \sum_i x_i^k)^l$ | $l$ | $k$ | - | - | $l$ | 0 | - | - | 1 | 0 | 1 | 0 | |
| min/max | $\min_i x_i / \max_i x_i$ | 0 | 1 | $1/r$ | $r$ | $1/s$ | $s$ | - | - | 1 | 1 | 1 | 0 | $r, s \to \infty$ |
| max/min | $\max_i x_i / \min_i x_i$ | $1/r$ | $r$ | - | - | 0 | 1 | $1/s$ | $s$ | 1 | 0 | 1 | 1 | $r, s \to \infty$ |

Table 1: Different functions achievable by varying the parameters in the formulation in Eq. 2

- LAF performs comparably to PNA on random graph generation tasks, outperforming several graph neural networks architectures including GAT (Veličković et al., 2018) and GIN (Xu et al., 2019)

The rest of this work is structured as follows. In Section 2 we define the LAF framework and show how appropriate parametrizations of LAF allow to represent a wide range of popular aggregation functions. In Section 3 we discuss some relevant related work. Section 4 reports synthetic and real-world experiments showing the advantages of LAF over (sets of) predifined aggregators. Finally, conclusions and pointers to future work are discussed in Section 5.

## 2 THE LEARNING AGGREGATION FUNCTION FRAMEWORK

We use $\boldsymbol{x} = \{x_1, \ldots, x_N\}$ to denote finite multisets of real numbers $x_i \in \mathbb{R}$. Note that directly taking $\boldsymbol{x}$ to be a multiset, not a vector, means that there is no need to define properties like exchangeability or permutation equivariance for operations on $\boldsymbol{x}$. An aggregation function *agg* is any function that returns for any multiset $\boldsymbol{x}$ of arbitrary cardinality $N \in \mathbb{N}$ a value $agg(\boldsymbol{x}) \in \mathbb{R}$.

Standard aggregation functions like *mean* and *max* can be understood as (normalized) $L_p$-norms. We therefore build our parametric LAF aggregator around generalized $L_p$-norms of the form

$$L_{a,b}(\boldsymbol{x}) := \left( \sum_i x_i^b \right)^a \qquad (a, b \geq 0). \tag{1}$$

$L_{a,b}$ is invariant under the addition of zeros: $L_{a,b}(\boldsymbol{x}) = L_{a,b}(\boldsymbol{x} \cup \boldsymbol{0})$ where $\boldsymbol{0}$ is a multiset of zeros of arbitrary cardinality. In order to also enable aggregations that can represent *conjunctive* behavior such as *min*, we make symmetric use of aggregators of the multisets $\boldsymbol{1} - \boldsymbol{x} := \{1 - x_i | x_i \in \boldsymbol{x}\}$. For $L_{a,b}(\boldsymbol{1} - \boldsymbol{x})$ to be a well-behaved, dual version of $L_{a,b}(\boldsymbol{x})$, the values in $\boldsymbol{x}$ need to lie in the range $[0, 1]$. We therefore restrict the following definition of our *learnable aggregation function* to sets $\boldsymbol{x}$ whose elements are in $[0, 1]$:

$$\mathrm{LAF}(\boldsymbol{x}) := \frac{\alpha L_{a,b}(\boldsymbol{x}) + \beta L_{c,d}(\boldsymbol{1} - \boldsymbol{x})}{\gamma L_{e,f}(\boldsymbol{x}) + \delta L_{g,h}(\boldsymbol{1} - \boldsymbol{x})} \tag{2}$$

defined by tunable parameters $a, \ldots, h \geq 0$, and $\alpha, \ldots, \delta \in \mathbb{R}$. In cases where sets need to be aggregated whose elements are not already bounded by 0, 1, we apply a sigmoid function to the set elements prior to aggregation.

Table 1 shows how a number of important aggregation functions are special cases of LAF (for values in $[0, 1]$). We make repeated use of the fact that $L_{0,1}$ returns the constant 1. For max and min LAF only provides an asymptotic approximation in the limit of specific function parameters (as indicated in the limits column of Table 1). In most cases, the parameterization of LAF for the functions in Table 1 will not be unique. Being able to encode the powers of moments implies that e.g. the variance of $\boldsymbol{x}$ can be expressed as the difference $1/N \sum_i x_i^2 - (1/N \sum_i x_i)^2$ of two LAF aggregators.

Since LAF includes sum-aggregation, we can adapt the results of Zaheer et al. (2017) and Wagstaff et al. (2019) on the theoretical universality of sum-aggregation as follows.

**Proposition 1** *Let $\mathcal{X} \subset \mathbb{R}$ be countable, and $f$ a function defined on finite multisets with elements from $\mathcal{X}$. Then there exist functions $\phi : \mathcal{X} \to [0, 1]$, $\rho : \mathbb{R} \to \mathbb{R}$, and a parameterization of* LAF, *such that $f(\boldsymbol{x}) = \rho(LAF(\phi\boldsymbol{x}); \alpha, \beta, \gamma, \delta, a, b, c, d)$, where $\phi\boldsymbol{x}$ is the multiset $\{\phi(x) | x \in \boldsymbol{x}\}$.*

A proof in Wagstaff et al. (2019) for a very similar proposition used a mapping from $\mathcal{X}$ into the reals. Our requirement that LAF inputs must be in $[0, 1]$ requires a modification of the proof (contained in the supplementary material), which for the definition of $\phi$ relies on a randomized construction. Proposition 1 shows that we retain the theoretical universality guarantees of Zaheer et al. (2017), while enabling a wider range of solutions based on continuous encoding and decoding functions.

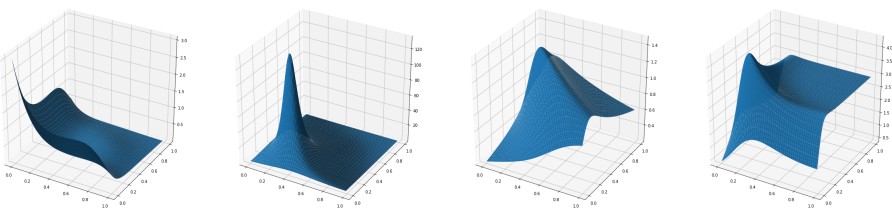

Figure 1: LAF functions with randomly generated parameters

It should be emphasized at this point that the primary purpose of LAF is not to provide a uniform representation of different standard aggregators as displayed in Table 1, but to enable a continuum of intermediate and hybrid aggregators. Figure 1 shows the graphs of 4 different randomly generated LAF functions over the unit square $[0, 1] \times [0, 1]$, i.e., evaluated over sets of size 2. Parameters $\alpha, \ldots, \gamma$ were randomly sampled in the interval $[0, 1]$; parameters $b, d, f, h$ are randomly sampled from the integers $0, \ldots, 5$, and $a, c, e, g$ are obtained as $1/i$ with $i$ a random integer from $0, \ldots, 5$. The figure illustrates the rich repertoire of aggregation functions with different qualitative behaviors already for non-extreme parameter values.

## 2.1 LAF ARCHITECTURE

LAF can be easily used as a module of a larger architecture suitable for learning on sets. Several LAF units can be combined as shown in Figure 2, to capture different aspects of the input set, which can be in general a set of vectors $\boldsymbol{x} = \{x_1, \ldots, x_N\}$ where $x_i \in \mathbb{R}^d$. Note that multiple aggregators are also used in related frameworks such as DeepSets (Zaheer et al., 2017) or Graph Neural Networks (Veličković et al., 2018; Corso et al., 2020). A module with $r$ LAF units takes as input $d$-dimensional vectors and produces a vector of size $r \times d$ as output. Each LAF unit performs an *element-wise* aggregation of the vectors in the set such that $L_{k,j} = \text{LAF}(\{x_{i,j}, \ldots, x_{N,j}\}; \alpha_k, \beta_k, \gamma_k, \delta_k, a_k, b_k, c_k, d_k)$ for $k = 1, \ldots, r$ and $j = 1, \ldots, d$. The output vector can be then fed into the next layer.

## 3 RELATED WORK

Several studies address the problem of aggregating data over sets. Sum-decomposition strategies have been used in (Zaheer et al., 2017) for points cloud classification and set expansion tasks and in (Santoro et al., 2017) for question answering and dynamic physical systems computation. Max, sum and average are standard aggregation functions for node neighborhoods in graph neural networks (Hamilton et al., 2017; Kipf & Welling, 2017; Xu et al., 2019; Veličković et al., 2018). Zaheer et al. (2017) first proved universal representation results for these standard aggregators when combined with learned mappings over inputs and results of the aggregation. However, Wagstaff et al. (2019) showed that these universality results are of little practical use, as they either require highly discontinuous mappings that would be extremely difficult to learn, or a latent dimension that is at least the size of the maximum number of input elements.

*Uninorms* (Yager & Rybalov, 1996) are a class of aggregation functions in fuzzy logic that can behave in a *conjunctive*, *disjunctive* or *averaging* manner depending on a parameter called *neutral element*. Melnikov & Hüllermeier (2016) proposed to learn fuzzy aggregators by adjusting these learnable parameters, showing promising results on combining reviewers scores on papers into an

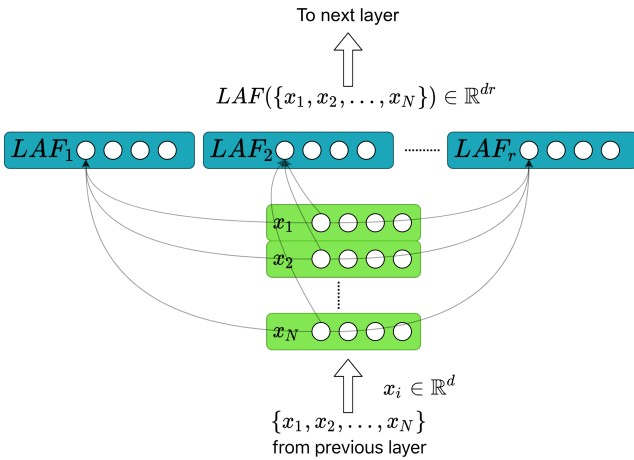

Figure 2: End-to-end LAF architecture.

overall decision of acceptance or reject. Despite the advantage of incorporating different behaviours in one single function, uninorms present discontinuities in the regions between aggregators, making them not amenable to be utilized in fully differentiable frameworks. Furthermore the range of possible behaviours is restricted to those commonly used in the context of fuzzy-logic.

The need for considering multiple candidate aggregators is advocated in a very recent work that was developed in parallel with our framework (Corso et al., 2020). The resulting architecture, termed *Principal Neighborhood Aggregation* (PNA) combines multiple standard aggregators, including most of the ones we consider in the LAF framework, adjusting their outputs with degree scalers. However, the underlying philosophy is rather different. PNA aims at learning to select the appropriate aggregator(s) from a pool of candidates, while LAF explores a continuous space of aggregators that includes standard ones as extreme cases. Our experimental evaluation shows that PNA has troubles in learning aggregators that generalize over set sizes, despite having them in the pool of candidates, likely because of the quasi-combinatorial structure of its search space. On the other hand, LAF can successfully learn even the higher moment aggregators and consistently outperforms PNA.

Closely connected, but somewhat complementary to aggregation operators are *attention mechanisms* (Bahdanau et al., 2015; Vaswani et al., 2017). They have been explored to manipulate set data in Lee et al. (2019) and in the context of multi-instance learning (Ilse et al., 2018). Attention operates at the level of set elements, and aims at a transformation (weighting) of their representations such as to optimize a subsequent weighted sum-aggregation. While the objectives of attention-based frameworks and LAF partially overlap, they are functionally quite different. Exploring combinations of LAF with attention mechanisms is a possible subject of future work.

## 4 EXPERIMENTS

In this section, we present and discuss experimental results showing the potential of the LAF framework on both synthetic and real-world tasks[1]. Synthetic experiments are aimed at showing the ability of LAF to learn a wide range of aggregators and its ability to generalize over set sizes (i.e., having test-set sets whose cardinality exceeds the cardinality of the training-set sets), something that alternative architectures based on predefined aggregators fail to achieve. We use DeepSets, PNA, and LSTM as representatives of these architectures. The LSTM architecture corresponds to a version of DeepSets where the aggregation function is replaced by a LSTM layer. Experiments on diverse tasks including point cloud classification, text concept set retrieval and graph properties prediction are aimed at showing the potential of the framework on real-world applications.

---

[1]The source code is now available in the supplementary material

## 4.1 Experiments on Scalars

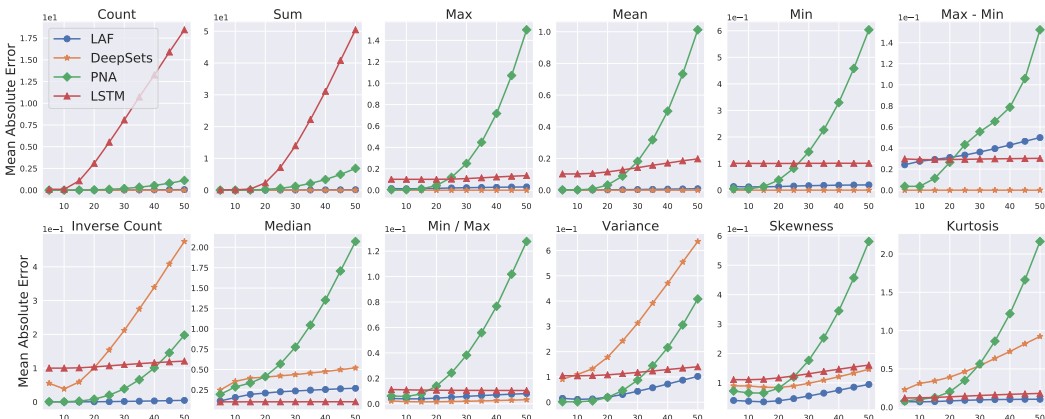

Figure 3: Test performances for the synthetic experiment with integer scalars on increasing test set size. The x axis of the figures represents the maximum test set cardinality, whereas the y axis depicts the MAE error. The dot, star, diamond and triangle symbols denote LAF, DeepSets, PNA, and LSTM respectively.

This section shows the learning capacity of the LAF framework to learn simple and complex aggregation functions where constituents of the sets are simple numerical values. In this setting we consider sets made of scalar integer values. The training set is constructed as follows: for each set, we initially sample its cardinality $K$ from a uniform distribution taking values in $\{2, M\}$, and then we uniformly sample $K$ integers in $0, \ldots, 9$. For the training set we use $M = 10$. We construct several test sets for different values of $M$ ($M = 5, 10, 15, 20, 25, 30, 35, 40, 45, 50$). This implies that models need to generalize to larger set sizes. Contrarily to the training set, each test set is constructed in order to diversify the target labels it contains, so as to avoid degenerate behaviours for large set sizes (e.g., maximum constantly equal to 9). Each synthetic dataset is composed of 100,000 sets for training, 20,000 set for validating and 100,000 for testing.

The number of aggregation units is set as follows. The model contains nine LAF (Equation 2) units, whose parameters $\{a_k, \ldots, h_k\}$, $k = 1, \ldots, 9$ are initialized according to a uniform sampling in $[0, 1]$ as those parameters must be positive, whereas the coefficients $\{\alpha, \ldots, \delta\}$ are initialized with a Gaussian distribution with zero mean and standard deviation of 0.01 to cover also negative values. The positivity constraint for parameters $\{a, b, ..., h\}$ is enforced by projection during the optimization process. The remaining parameters can take on negative values. DeepSets also uses nine units: three max units, three sum units, and three mean units and PNA uses seven units: mean, max, sum, standard deviation, variance, skewness and kurtosis. Preliminary experiments showed that expanding the set of aggregators for PNA with higher order moments only leads to worse performance. Each set of integers is fed into an embedding layer (followed by a sigmoid) before performing the aggregation function. DeepSets and PNA do need an embedding layer (otherwise they would have no parameters to be tuned). Although LAF does not need an embedding layer, we used it in all models to make the comparison more uniform. The architecture details are reported in the supplementary material. We use the Mean Absolute Error (MAE) as a loss function to calculate the prediction error.

Figure 3 shows the trend of the MAE error for the three methods for increasing test set sizes, for different types of target aggregators. As expected, DeepSets manages to learn the identity function and thus correctly models aggregators like sum, max and mean. Even if LAF needs to adjust its parameters in order to properly aggregate the data, its performance are competitive with those of DeepSets. When moving to more complex aggregators like inverse count, median or moments of different orders, DeepSets fails to learn the latent representation. One the other hand, the performance of LAF is very stable for growing set sizes. While having in principle at its disposal most of the target aggregators (including higher order moment) PNA badly overfits over the cardinality of sets in the training set in all cases (remember that the training set contains sets of cardinality at most 10). The reason why LAF substantially outperforms PNA on large set sizes could be explained in terms of a greater flexibility to adapt to the learnt representation. Indeed, LAF parameters can

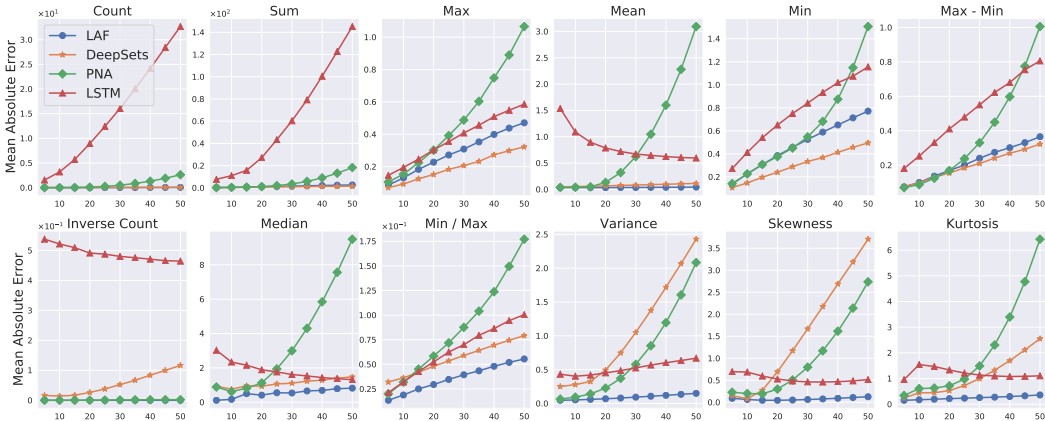

Figure 4: Test performances for the synthetic experiment on MNIST digits on increasing test set size. The x axis of the figures represents the maximum test set cardinality, whereas the y axis depicts the MAE error. The dot, star, diamond and traingle symbols denote LAF, DeepSets, PNA and LSTM respectively.

adjust the *laf* function to be compliant with the latent representation even if the input mapping fails to learn the identity. On the other hand, having a bunch of fixed, hard-coded aggregators, PNA needs to be able to both learn the identity mapping and select the correct aggregator among the candidates. Finally, LSTM exhibits generally poor results when compared to the other methods, particularly in the case of the count and the sum.

## 4.2 MNIST DIGITS

In this section, we modify the previous experimental setting to process MNIST images of digits. The dataset is the same as in the experiment on scalars, but integers are replaced by randomly sampling MNIST images for the same digits. Instances for the training and test sets are drawn from the MNIST training and test sets, respectively. This experiment aims to demonstrate the ability of LAF to learn from more complex representations of the data by plugging it into end-to-end differentiable architectures. Contrarily to the model of the previous section, here we use three dense layers for learning picture representations before performing the aggregation function. The architecture details are reported in the supplementary material.

Figure 4 shows the comparison of LAF, DeepSets, PNA, and LSTM in this setting. Results are quite similar to those achieved in the scalar setting, indicating that LAF is capable of effectively backpropagating information so as to drive the learning of an appropriate latent representation, while DeepSets, PNA, and LSTM suffer from the same problems seen in aggregating scalars.

Furthermore, Figure 5 provides a qualitative evaluation of the predictions of the LAF, DeepSets, and PNA methods on a representative subset of the target aggregators. The images illustrate the correlation between the true labels and the predictions. LAF predictions are distributed over the diagonal line, with no clear bias. On the other hand, DeepSets and PNA perform generally worse than LAF, exhibiting higher variances. In particular, for inverse count and kurtosis, DeepSets and PNA predictions are condensed in a specific area, suggesting an overfitting on the training set.

## 4.3 POINT CLOUD

In order to evaluate LAF on real-world dataset, we consider point cloud classification, a prototype task for set-wise prediction. Therefore, we run experimental comparisons on the ModelNet40 (Wu et al., 2015) dataset, which consists of 9,843 training and 2,468 test point clouds of objects distributed over 40 classes. The dataset is preprocessed following the same procedure described by Zaheer et al. (2017). We create point clouds of 100 and 1,000 three-dimensional points by adopting the point-cloud library's sampling routine developed by Rusu & Cousins (2011) and normalizing each set of points to have zero mean (along each axis) and unit (global) variance. We refer with

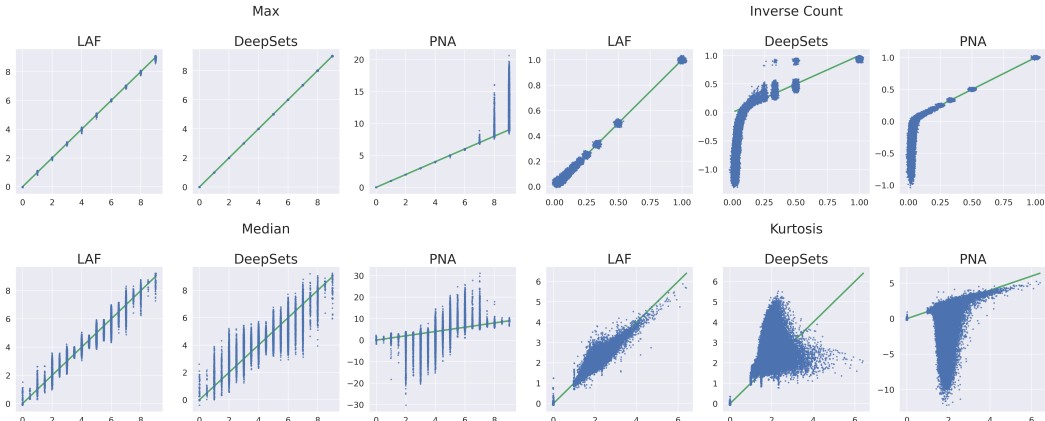

Figure 5: Scatter plots of the MNIST experiment comparing true (x axis) and predicted (y axis) values with 50 as maximum test set size. The target aggregations are *max* (up-left), *inverse count* (up-right), *median* (bottom-left) and *kurtosis* (bottom-right).

P100 and P1000 to the two datasets. For all the settings, we consider the same architecture and hyper-parameters of the DeepSets permutation invariant model described by Zaheer et al. (2017). For LAF, we replace the original aggregation function (max) used in DeepSets with 10 LAF units, while for PNA we use the concatenation of max, min, mean, and standard deviation, as proposed by the authors. For PNA we do not consider any scaler, as the cardinalities of the sets are fixed.

Results in Table 2 show that LAF produces an advantage in the lower resolution dataset (i.e. on P100), while it obtains comparable (and slightly more stable) performances in the higher resolution one (i.e. on P1000). These results suggest that having predefined aggregators is not necessarily an optimal choice in real world cases, and that the flexibility of LAF in modeling diverse aggregation functions can boost performance and stability.

Table 2: Results on the Point Cloud classification task. Accuracies with standard deviations (calculated on 5 runs) for the ModelNet40 dataset.

| METHOD | P100 | P1000 |
|---|---|---|
| DEEPSETS | 82.0±2.0% | 87.0±1.0% |
| PNA | 82.9±0.7% | 86.4±0.6% |
| LSTM | 78.7±1.1% | 82.2±1.7% |
| LAF | **84.0±0.6%** | **87.0±0.5%** |

### 4.4 SET EXPANSION

Following the experimental setup of DeepSets, we also considered the *Set Expansion* task. In this task the aim is to augment a set of objects of the same class with other similar objects, as explained in (Zaheer et al., 2017). The model learns to predict a score for an object given a query set and decide whether to add the object to the existing set. Specifically, Zaheer et al. (2017) consider the specific application of set expansion to text concept retrieval. The idea is to retrieve words that belong to a particular concept, giving as input set a set of words having the same concept. We employ the same model and hyper-parameters of the original publication, where we replace the sum-decomposition aggregation with LAF units for our methods and the min, max, mean, and standard deviation aggregators for PNA.

We trained our model on sets constructed from a vocabulary of different size, namely *LDA-1K*, *LDA-3K* and *LDA-5K*. Table 3 shows the results of LAF, DeepSets and PNA on different evaluation metrics. We report the retrieval metrics recall@K, median rank and mean reciprocal rank. We also report the results on other methods the authors compared to in the original paper. More details on the

Table 3: Results on Text Concept Set Retrieval on LDA-1k, LDA-3k, and LDA-5k. Bold values denote the best performance for each metric.

| METHOD | LDA-1$k$ (VOCAB = 17$k$) RECALL(%) | | | | | LDA-3$k$ (VOCAB = 38$k$) RECALL(%) | | | | | LDA-5$k$ (VOCAB = 61$k$) RECALL(%) | | | | |
|---|---|---|---|---|---|---|---|---|---|---|---|---|---|---|---|
| | @10 | @100 | @1K | MRR | MED. | @10 | @100 | @1K | MRR | MED. | @10 | @100 | @1K | MRR | MED. |
| RANDOM | 0.06 | 0.6 | 5.9 | 0.001 | 8520 | 0.02 | 0.2 | 2.6 | 0.000 | 28635 | 0.01 | 0.2 | 1.6 | 0.000 | 30600 |
| BAYES SET | 1.69 | 11.9 | 37.2 | 0.007 | 2848 | 2.01 | 14.5 | 36.5 | 0.008 | 3234 | 1.75 | 12.5 | 34.5 | 0.007 | 3590 |
| W2V NEAR | 6.00 | **28.1** | 54.7 | 0.021 | 641 | 4.80 | 21.2 | 43.2 | 0.016 | 2054 | 4.03 | 16.7 | 35.2 | 0.013 | 6900 |
| NN-MAX | 4.78 | 22.5 | 53.1 | 0.023 | 779 | 5.30 | 24.9 | 54.8 | 0.025 | 672 | 4.72 | 21.4 | 47.0 | 0.022 | 1320 |
| NN-SUM-CON | 4.58 | 19.8 | 48.5 | 0.021 | 1110 | 5.81 | 27.2 | 60.0 | 0.027 | 453 | 4.87 | 23.5 | 53.9 | 0.022 | 731 |
| NN-MAX-CON | 3.36 | 16.9 | 46.6 | 0.018 | 1250 | 5.61 | 25.7 | 57.5 | 0.026 | 570 | 4.72 | 22.0 | 51.8 | 0.022 | 877 |
| DEEPSETS | 5.53 | 24.2 | 54.3 | 0.025 | 696 | 6.04 | 28.5 | 60.7 | 0.027 | 426 | 5.54 | 26.1 | 55.5 | 0.026 | 616 |
| DEEPSETS* | 5.89 | 26.0 | **55.3** | 0.026 | **619** | 7.56 | 28.5 | **64.0** | 0.035 | 349 | 6.49 | 27.9 | **56.9** | 0.030 | 536 |
| PNA | 5.56 | 24.7 | 53.2 | 0.027 | 753 | 7.04 | 27.2 | 58.7 | 0.028 | 502 | 5.47 | 23.8 | 52.4 | 0.025 | 807 |
| LSTM | 4.29 | 21.5 | 52.6 | 0.022 | 690 | 5.56 | 25.7 | 58.8 | 0.026 | 830 | 4.87 | 23.8 | 55.0 | 0.022 | 672 |
| LAF | **6.51** | 26.6 | 54.5 | **0.030** | 650 | **8.14** | **32.3** | 62.8 | **0.037** | 339 | **6.71** | **28.3** | **56.9** | **0.031** | 523 |

other methods in the table can be found in the original publication. Briefly, *Random* samples a word uniformly from the vocabulary; *Bayes Set* (Ghahramani & Heller, 2006); *w2v-Near* computes the nearest neighbors in the word2vec (Mikolov et al., 2013) space; *NN-max* uses a similar architecture as our DeepSets but uses max pooling to compute the set feature, as opposed to sum pooling; *NN-max-con* uses max pooling on set elements but concatenates this pooled representation with that of query for a final set feature; *NN-sum-con* is similar to NN-max-con but uses sum pooling followed by concatenation with query representation. For the sake of fairness, we have rerun DeepSets using the current implementation from the authors (indicated as DeepSet* in Table 3), exhibiting better results than the ones reported in the original paper. Nonetheless, LAF outperforms all other methods in most cases, especially on *LDA-3K* and *LDA-5K*.

## 4.5 MULTI-TASK GRAPH PROPERTIES

Corso et al. (2020) defines a benchmark consisting of 6 classical graph theory tasks on artificially generated graphs from a wide range of popular graph types like Erdos-Renyi, Barabasi-Albert or star-shaped graphs. Three of the tasks are defined for nodes, while the other three for whole graphs. The node tasks are the single-source shortest-path lengths (N1), the eccentricity (N2) and the Laplacian features (N3). The graph tasks are graph connectivity (G1), diameter (G2), and the spectral radius (G3). For more details about the experimental settings please refer to Corso et al. (2020).

Table 4: Results on the Multi-task graph properties prediction benchmark. Results are expressed in log 10 of mean squared error.

| METHOD | N1 | N2 | N3 | G1 | G2 | G3 |
|---|---|---|---|---|---|---|
| BASELINE | -1.87 | -1.50 | -1.60 | -0.62 | -1.30 | -1.41 |
| GIN | -2.00 | -1.90 | -1.60 | -1.61 | -2.17 | -2.66 |
| GCN | -2.16 | -1.89 | -1.60 | -1.69 | -2.14 | -2.79 |
| GAT | -2.34 | -2.09 | -1.60 | -2.44 | -2.40 | -2.70 |
| MPNN (MAX) | -2.33 | -2.26 | -2.37 | -1.82 | -2.69 | -3.52 |
| MPNN (SUM) | -2.36 | -2.16 | -2.59 | -2.54 | -2.67 | -2.87 |
| PNA (NO SCALERS) | -2.54 | -2.42 | -2.94 | **-2.61** | -2.82 | -3.29 |
| PNA | **-2.89** | **-2.89** | **-3.77** | **-2.61** | **-3.04** | -3.57 |
| LAF | -2.13 | -2.20 | -1.67 | -2.35 | -2.77 | **-3.63** |

We compare LAF against PNA by simply replacing the original PNA aggregators and scalers with 100 LAF units (see Equation 2). Table 4 shows that albeit these datasets were designed to highlight the features of the PNA architecture, that outperforms a wide range of alternative graph neural network approaches LAF produces competitive results, outperforming state-of-the-art GNN approaches like GIN (Xu et al., 2019), GCN (Kipf & Welling, 2017) and GAT (Veličković et al., 2018) and even improving over PNA on spectral radius prediction.

## 5 CONCLUSIONS

The theoretical underpinnings for sum aggregation as a universal framework for defining set functions do not necessarily provide a template for practical solutions. Therefore we introduced LAF, a framework for learning aggregation functions that make use of a parametric aggregator to effectively explore a rich space of possible aggregations. LAF defines a new class of aggregation functions, which include as special cases widely used aggregators, and also has the ability to learn complex functions such as higher-order moments. We empirically showed the generalization ability of our method on synthetic settings as well as real-world datasets, providing comparisons with state-of-the-art sum-decomposition approaches and recently introduced techniques. The flexibility of our model is a crucial aspect for potential practical use in many deep learning architectures, due to its ability to be easily plugged into and learned in end-to-end architectures. The portability of LAF opens a new range of possible applications for aggregation functions in machine learning methods, and future research in this direction can enhance the expressivity of many architectures and models that deal with unstructured data.

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
