# OpenReview forum: "Learning Aggregation Functions"
_ICLR.cc/2021/Conference — Reject_

### Official Review · AnonReviewer3 · 2020-10-25
**Learning Aggregation Functions over sets.**

**Rating:** 5
**Confidence:** 3

**Review:**

The paper proposes a rational approximation approach for learning aggregation functions. In general the paper is well written and technically sound, although there are some questions that could help the readers better understand the paper.

One of the problems with learnable rational approximations is the potential of finding a pole, e.g., x/0, you do not mention how you avoid/use? such a situation in your approach, whether this causes instabilities during learning, etc. Could you mention something about this?

Regarding the use of the sigmoid to transform the values of x to the range [0,1], it is not clear to me, how you can recover the mean with such low error, i.e., how can you achieve $\mu(x) ~=  (\sum sigmoid(x)^b)^a / N$. Without the sigmoid transformation it is clear as presented in table 1. But with the sigmoid transformation is not as straightforward to see, although you show very good results in Fig. 2. Furthermore the sigmoid transformation destroys the linearity needed when the values of x are larger. Could you also give some intuition on why using the sigmoid is superior compared to a minmax scaling process?

You mention that you use 9 LAF(x) aggregation functions in the experiments, could you explain more what do you mean exactly? are you using a mixture of LAF(x) models? are they independent of each other and train for the different target functions?

In general, I find the paper interesting and the results promising, just a bit more explanation could help the reader understand the benefits and intuition behind the decisions.

---

> ### Author Response · Authors · 2020-11-17
> **Response to Reviewer 3**
>
>  - One of the problems with learnable rational approximations is the
>     potential of finding a pole, e.g., x/0, you do not mention how you
>     avoid/use? such a situation in your approach, whether this causes
>     instabilities during learning, etc. Could you mention something
>     about this?
>
>     This is indeed a problem when LAF is applied in
>     practice. Our strategy to avoid the problem is simply a clipping of
>     the LAF denominator to a small signed constant $\epsilon$ if it is
>     lower (in absolute value) than $\epsilon$.
>
> -   Regarding the use of the sigmoid to transform the values of x to the
>     range \[0,1\], it is not clear to me, how you can recover the mean
>     with such low error, i.e., how can you achieve
>     $\mu(x) ~= (\sum sigmoid(x)^b)^a / N$. Without the sigmoid
>     transformation it is clear as presented in table 1. But with the
>     sigmoid transformation is not as straightforward to see, although
>     you show very good results in Fig. 2. Furthermore the sigmoid
>     transformation destroys the linearity needed when the values of x
>     are larger. Could you also give some intuition on why using the
>     sigmoid is superior compared to a minmax scaling process?
>
>     We apologize as there was a mistake in the text. The
>     architecture reported in Section 4.1 was the one used for the
>     experiments of Section 4.2. The architecture for experiments in Sec
> 4.1 is indeed much simpler: $$\begin{aligned}
>     & & \mbox{INPUT: } \\{x_1, \ldots, x_N\\}, x_i \in \\{0,1,\ldots, 9\\} \Rightarrow \\\\
>     & & \mbox{Embedding } (10 \times 10) \Rightarrow \mbox{ Sigmoid }
>     \Rightarrow 9 \mbox{ LAF aggregatators } \Rightarrow \\\\
>     & & \mbox{Linear layer } (9 \times 1)
>     \end{aligned}$$
>     We have clarified the architecture in the text
>     and add the full details in the Supplementary Material. Furthermore,
>     we have also expanded Section 2 with a subsection describing
>     explicitly the end-to-end architecture of our approach. In this
>     setting, LAF does not operate directly on the integers numbers but
>     on a representation of them. The sigmoid function is applied
>     elementwise to the representations. Consider an example where the
>     target function is the mean and inputs are sets of integers taking
>     values in $\{0,\ldots,9\}$. A simple learnable representation for
>     the integers is the one-hot vectors. Let's also assume to use only a
>     LAF unit which learns exactly the parameters for the mean (e.g. the
>     one reported in Table 1). The output of the LAF is a vector of size
>     10, so we add an extra linear layer to map it to a single scalar. To
>     further clarify we report a simple case: Given an input set of 3
>     digits whose labels are $\{3, 1, 1\}$, the learnt features are
>     $(0,0,0,1,0,0,0,0,0,0)$, $(0,1,0,0,0,0,0,0,0,0)$,
>     $(0,1,0,0,0,0,0,0,0,0)$. Those three features would be the input for
>     LAF which in turn outputs $(0,2/3,0,1/3,0,0,0,0,0,0)$. The weights
>     of the linear layer after LAF needs to be $(0,1,2,3,4,5,6,7,8,9)$,
>     so to obtain
>     $(0,2/3,0,1/3,0,0,0,0,0,0)^T(0,1,2,3,4,5,6,7,8,9) = 1.667$. Nothing
>     prevents us from considering different rescaling functions such as
>     min-max rescaling, and indeed this was our initial choice (with min
>     and max estimated batch-wise). However, in practise we found out
>     that it performed worse than the sigmoid, possibly because of the
>     approximation due to the batch-wise estimation of min and max
>     values.
>
> -   You mention that you use 9 LAF(x) aggregation functions in the
>     experiments, could you explain more what do you mean exactly? are
>     you using a mixture of LAF(x) models? are they independent of each
>     other and train for the different target functions?
>
>     Indeed. We create 9 LAF independently and then we concatenate them. On top
>     of that there is a linear layer (or more generally, a second neural
>     network) that maps the concatenated LAF outputs into the desired
>     output dimension. We have improved the description and clarification
>     of the whole architecture in Section 2.1.

---

### Official Review · AnonReviewer4 · 2020-10-27
**A novel method for aggregating the information from sets**

**Rating:** 6
**Confidence:** 3

**Review:**

Summary:

Universal function representation guarantee requires either highly discontinuous mappings or a highly dimensional latent space. For this reason the authors propose a new parametric family of aggregation functions, called LAF (for learning aggregation functions). It can be seen as a smooth version of the class of functions that are shown in DeepSets. LAF aggregator could learn all standard aggregation functions. Moreover in experiments the autors shows that LAF surpasses other aggregation methods.

=============================================================================

Pros:

1. The authors shows, that all standard aggregation functions are achievable by varying the parameters in the formulation of LAF. Moreover LAF enables a neural network to use a continuum of intermediate and hybrid aggregators.

2. Comprehensive ablation study that compares LAF to DeepSets and PNA on digits and MNIST images. In the study the goal is to learn a different types of target aggregation. The results shows that LAF could learn all the given types of aggregation methods as well as it could generalize well to the size of the test set (and thus is not overfitting to the size of the training set as the other methods).

3. The authors provide an extensive set of experiments on a wide range of datasets, including point clouds, set expansion and graph properties. On most of the given tasks LAF is superior to other methods.

=============================================================================

Cons:

Not all the details about LAF aggregation are clear to me. The authors should consider rewriting a section 2 (with a description of the aggregation), considering the points I list below.

1. In the manuscript the autors state that LAF is using the tunable parameters a,...,h and alpha,...,delta, however they do not show how to initialize these parameters and do not tell whether the model is sensitive to these values.

2. The autors state that tunable parameters a,...,h are greater or equal than zero. However hey do not show how to achieve this condition. Whether they use exponent, non-linearity or do it in another way.

3. In the definition of LAF aggregator, the authors states that x should be a real number, however looking at the experiments, it seems to me that it could also be applied to vectors. Please correct me if I'm wrong, but if I am right, than please answer the question whether in this situation a,...,h,alpha,...,delta are still the scalars or whether are they vecors?

4. The more detailed description of builded networks should be included (even in the appendix). It is not clear to me, how the authors made thir networks. E.g. in section 4.1 they state that 'The LAF model contains nine LAF(x) aggregation functions' - does this mean that in the final aggregation layer you create 9 independently working LAF aggregators and then concatenate them and pass to final prediction layer? If yes, then what in the situation with aggregating the informations from vectors? Do you still make concatenaction?

=============================================================================

Questions during rebuttal period:

1. Why section 5 (Multi-task graph properties) is not the sub-section of section 4 (Experiments)?

2. Usually using a sigmoid could disturb the training of neural network. Could you create the experiment (on a real dataset), where you delete the sigmoid as well as parts with 1-x form the LAF aggregator?

3. Could you create some experiments with LAF as the final aggregator for neural networks? More specific, for exampe, could you use LAF instead of mean average pooling in image classification using ResNet or some text classification task?

=============================================================================

=============================================================================

Reasons for score:

I vote for accepting this paper. The idea proposed by the authors is novel and elegant. Moreover experiments shows that the proposed model is superior to the other models with whom it has been compared. My major concern is about the clarity of the paper. Hopefully the authors can address my concern in the rebuttal period.

---

> ### Author Response · Authors · 2020-11-17
> **Response to Reviewer 4**
>
> -   In the manuscript the authors state that LAF is using the tunable
>     parameters a,...,h and alpha,...,delta, however they do not show
>     how to initialize these parameters and do not tell whether the model
>     is sensitive to these values.
>     Sorry for that. We have now explicitly mentioned the initialization strategy in the paper. The
>     parameters are initialized with a simple strategy. Particularly, the
>     parameters $\\{a,b,...,h\\}$ are initialized with a uniform sampling
>     in $[0,1]$ as those parameters must be positive, whereas the
>     coefficients $\\{\alpha, \beta, \gamma, \delta \\}$ are initialized
>     with a gaussian distribution centered at zero with a variance of
>     0.01 to cover also negative values. We have also tried to initialize
>     the weights all uniform in $[0,1]$ but we did not obtain any
>     significant improvement nor deterioration. We didn't experience any
>     sensitivity due to the parameters inizialization when we use
>     multiple LAF units (see the analysis in the Supplementary Material).
>
> -   The authors state that tunable parameters a,...,h are greater or
>     equal than zero. However hey do not show how to achieve this
>     condition. Whether they use exponent, non-linearity or do it in
>     another way.
>     The positivity constraint for parameters
>     $\{a,b,...,h\}$ is enforced by projection during the optimization
>     process. The remaining parameters can take on negative values. We
>     have now explicitly mentioned the clipping strategy in the
>     paper.
>
> -   In the definition of LAF aggregator, the authors states that x
>     should be a real number, however looking at the experiments, it
>     seems to me that it could also be applied to vectors. Please correct
>     me if I'm wrong, but if I am right, than please answer the question
>     whether in this situation a,...,h,alpha,...,delta are still the
>     scalars or whether are they vectors?
>     The parameters for
>     one LAF unit are scalars but LAF can be applied to individual
>     components of sets of vectors. Figure 2 in the revised paper makes
>     this more clear.
>
> -   The more detailed description of built networks should be included
>     (even in the appendix).
>     You are totally right! We have now
>     explained the general architecture of our framework in Section 2.1
>     and we have reported in the Supplementary Material the details for
>     the whole experiments presented in the paper.
>
> -   It is not clear to me, how the authors made their networks. E.g. in
>     section 4.1 they state that 'The LAF model contains nine LAF(x)
>     aggregation functions' - does this mean that in the final
>     aggregation layer you create 9 independently working LAF aggregators
>     and then concatenate them and pass to final prediction layer? If
>     yes, then what in the situation with aggregating the informations
>     from vectors? Do you still make concatenation?
>     Indeed. We create 9 LAF independently and then we concatenate them. On top of
>     that there is a linear layer (or more generally, a second neural
>     network) that maps the concatenated LAF outputs into the desired
>     output dimension. We have improved the description and clarification
>     of the whole architecture in Section 2.1.
>
> -   Why section 5 (Multi-task graph properties) is not the sub-section
>     of section 4 (Experiments)?
>     Indeed. We apologize for that. It is now included in Section 4.
>
> -   Could you create some experiments with LAF as the final aggregator
>     for neural networks? More specific, for example, could you use LAF
>     instead of mean average pooling in image classification using ResNet
>     or some text classification task?
>     We could indeed use LAF as
>     final aggregator for neural networks. Nothing prevents us to use LAF
>     as an alternative trainable pooling layer after convolutions or even
>     as an alternative to e.g. MaxOut. In this paper we focused on
>     set-wise aggregation and thus our experiments show the use of LAF
>     inside architectures processing sets.

---

### Official Review · AnonReviewer2 · 2020-10-28
**I recommend to reject the paper, mainly due to unclear description of the method and the experiments. This needs to be improved before a proper review of the presented idea is possible. Also, comparison to more other methods should be performed.**

**Rating:** 3
**Confidence:** 5

**Review:**

Summary:
The paper proposes a parameterized learnable aggregation function (LAF) that can aggregate a multi-set of numbers (i.e. map them to a single real-valued number). This is different to prior works such as Deep Sets that use fixed aggregation functions such as max, mean, etc.

Strong Points:
- While Deep Sets have shown that is theoretically sufficient to have a sum aggregation, it is still unclear which kind of aggregation functions work well in practice. Hence, the paper addresses an interesting research question.
- The presented idea is rather simple, which I think is a strong point of the paper.
- I like the analysis in Table 1 that shows how different parameterizations of the LAF correspond to different functions such as sum, min, means, etc.
- I also like the evaluation presented in Figure 4.

Weak Points:
- The setup of the experiments with scalars is unclear. For example, a LAF is supposed to aggregate all scalars in the input. However, the paper states that the LAF model is comprised of 9 LAF functions. Why do we need 9 functions? And how are they composed into a single architecture? Similarly, the paper states that 'DeepSets contains three max units, three sum units, and three mean units'. However, a Deep Set should have one function mapping the input to an intermediate representation, a sum aggregation, and a function that maps the aggregation to the output. I don't see why the model needs three max, sum, and mean units.
- In experiment 1, it is reported that the 'input mapping is performed by three layers with the hyperbolic tangent as non-linear activation'. However, the input is simply a multi-set of scalars. It remains unclear to me why it can make sense to map scalars with a 3-layer network. Furthermore, a sigmoid is applied in case of LAF. Hence, it is unclear if the observed performance is due to the Sigmoid or due to the LAF. Furthermore, it is unclear why the paper uses tanh as activation function while Deep Set implementations use ReLUs. Also, the architecture of the output mapping is not described.
- Similar to experiment 1, the experimental setup in experiment 2 is unclear. Additionally, it is unclear how the aggregation of features (i.e. individual dimensions in MNIST images) is related to the aggregation function used to compute the target output of the multi-set.
- While the problem investigated in the paper is permutation-invariant, several methods have been proposed to approximate permutation invariant problems with recurrent architectures such as LSTMs and GRUs. However, no comparison to these kinds of methods is performed. A comparison would be interesting since they also learn an aggregation function.
- It would be interesting to see if the parameters of the LAF function are learned as expected, i.e. if they correlate with the expected values as listed in Table 1. An analysis of this question is missing.
- The paper does not share code or data to improve the reproducibility of the experiments.

---

> ### Author Response · Authors · 2020-11-17
> **Response to Reviewer 2**
>
> -The setup is unclear.
>  We apologize as there was a mistake in the text. The
> architecture reported in 4.1 was the one used for the
> experiments of 4.2. The architecture for experiments in
> 4.1 is indeed much simpler: $$\begin{aligned}
> & & \mbox{INPUT: } \{x_1, \ldots, x_N\}, x_i \in \{0,1,\ldots, 9\} \Rightarrow \\\\
> & & \mbox{Embedding } (10 \times 10) \Rightarrow \mbox{ Sigmoid }
> \Rightarrow 9 \mbox{ LAF aggregatators } \Rightarrow \\\\
> & & \mbox{Linear layer } (9 \times 1)
> \end{aligned}$$ The Embedding is needed for comparing to the
> other models which couldn't learn anything otherwise, as no
> parameters would have been included in the model. In principle for
> LAF one could completely remove the embedding and the method would
> still work as shown in Appendix B. We have clarified
> the architecture in the text and add the full details in the
> Appendix. Furthermore, we have also expanded Sec 2
> with a subsection describing explicitly the end-to-end architecture
> of our approach.
>
> -Why do we need 9 functions?
> to help optimization since the problem (even with just one LAF
> unit) is not convex. Distributing the result on several LAF
> units produces a smoother problem that is easier to optimize.
> The new Fig B.1 in Appendix B shows that more
> LAF units reduce the variance of the solution's quality.
> From the figure it is clear that sometimes
> with one LAF units we can obtain the desired solution. However,
> this would require a number of random restarts, which we can
> avoid by adding redundancy, i.e. more LAF
> units.
>
> -And how are they composed into a single architecture?
> We have improved the paper in Subsection 2.1 where we explicitly
> report the end-to-end architecture. In particular a whole set of
> vectors is fed as input into a first neural network, which
> outputs a representation for each instance in the set. On top of
> that, we aggregate the vectors element-wise with several LAF
> units and concatenate the outputs (similarly to DeepSets).
> Finally, we feed the concatenated LAF outputs into a second
> neural network. Formally, for $r$ LAF units we perform the
> following operation:\
> $L_{k,j} = LAF( \{x_{1,j}, \ldots x_{N,j} \} ; \{a_k, b_k, c_k, d_k, e_k, f_k, g_k, h_k, \alpha_k, \beta_k, \gamma_k, \delta_k\})$,
> where $k=1,\ldots, r$ and $j$ is the $j-$th component of the
> input vectors.
>
> -DeepSets contains three max units, three sum units...
> Preliminary comparison experiments exhibited that DeepSets with only one sum
> unit performed worse than DeepSets with three sum, mean, and max
> units. Furthermore, although the authors of DeepSets (Zaheer,
> et.al, 2017) explain the universality of the aggregation using
> sum-decomposition, it is clear in their code that they use also
> max and mean as aggregation functions. In particular, the best
> performance for the PointCloud dataset is achieved by the
> authors using max aggregators and tanh activations. Again, we
> decided to use three aggregation units per function because we
> experienced that redundancy helps and to have a more fair
> comparison with LAF which uses 9 aggregation functions
> too.
>
> -A sigmoid is applied in case of LAF...
> We used the Sigmoid for all methods being compared in Sec 4.
> Furthermore, please note that a
> single LAF unit which processes only scalar can actually recover the
> target aggregation function --- see the new Figure B.1 in the
> Supplementary Material.
>
> -Output mapping not described.
> We have improved the paper by describing the end-to-end architecture in Sec 2.1.
> The output mapping is simply a linear layer which maps the concatenated
> LAF units into a single scalar.
>
> -Setup of exp 2 unclear.
> We do not aggregate individual dimensions in MNIST
> images, rather we aggregate individual components of the
> representation vector constructed by previous layers. We have
> explicitly added the end-to-end structure of our model in Sec
> 2.1, which should clarify the whole architecture.
>
> -LSTMs and GRUs.
> For all the experiments in Sec 4 we have added LSTM,using
> different permutations of the sets in order to
> force the LSTM to be permutation invariant. Results show
> that LSTM in all the cases (even though providing reasonable
> results) performs worse than the other methods.
>
> -Parameters of LAF
> The same aggregation function can be realized with
> several assignments to LAF parameters; In Tab 1 we report one of
> these assignments but the learning process may actually find a
> different one. For example, consider the "count aggregator": in this
> case we have infinite solutions when $L_{a,b} = L_{c,d} = N$,
> $L_{e,f} = L_{g,h} =1$, and $\alpha + \beta = \gamma + \delta$.
> Similarly for the mean we have infinite solutions when
> $L_{a,b} = \sum$, $L_{e,f} = N$, $\beta = \delta = 0$, and
> $\alpha = \gamma$. The new appendix B shows a few examples
> reporting the values of the parameters found by the learning
> procedure when using a single LAF unit and several random restarts
> to ensure that a single LAF units does find the
> solution.
>
> Source code is now attached in the supplementary material.

---

### Official Review · AnonReviewer1 · 2020-10-28
**An attempt to learn aggregation functions with neural net compatibility**

**Rating:** 6
**Confidence:** 4

**Review:**

This paper addresses the problem of finding appropriate aggregation functions that can be used for instance in deep neural network architectures. One such function maps a variable length list of reals to a scalar.

The authors investigate the possibility to learn aggregation functions from data. To that end, they investigate an Lp norm based parametric model of aggregation functions called LAF that allows to learn a wide range of function including usual aggregation functions such as mean, max or min. They however restrict these functions to lists of reals that are in the unit interval.

The theoretical part of the paper is short. The authors simply show that their aggregation model has a rather high expressive power and invoke a theorem from prior arts to outline some universality guarantee (which could actually be explicitly recalled).

The limited amount of theory is compensated by extensive numerical experiments. The authors provide experimental evidence that their model generalizes fairly well to large sets of inputs. They also show that their model can be plugged to more conventional neural net layers and is backprop friendly.

This is an overall fair contribution which is well positioned as compared to prior arts. The main issue of this paper is, in my opinion, that the impact on the ML community will be limited if their LAF aggregation model does not become a gold standard like convolutional layers have become. To achieve such goal, one misses an application in which the LAF allows to bring disruptive results by leveraging features that cannot be derived through more usual architectures. In conjunction with this remark, the practical usability of LAF would be much wider if a generalization to sets of vectors would be proposed which could learn other features such as covariance.

Detailed Remarks:

Sec 2 : If b or e is equal to zero, then L_ab is not continuous. I think these should be limiting cases too.

Sec 4.1 : I fail to understand to what targets input sets are mapped to: one of the feature shown in Fig. 2 or a subset of them. Similarly, I also do not understand what is the global architecture of the approaches. It seems that have 9-dimensional outputs. Please clarify this.

Maybe the plots would be more informative if the y-axis was in log scale.

Sec 4.2 : Do you use convolutional layers before the aggregation layer ? Are they trained together ?

Sec 4.3 : Can you be more specific on the dataset pre-processing ? Why not directly use the raw dataset ?

---

> ### Author Response · Authors · 2020-11-17
> **Response to Reviewer 1**
>
>
> -   ...theorem from prior arts to outline some universality
>     guarantee...(which could actually be explicitly recalled).
>
>     We have explicitly stated and proven the universality guarantee in a
>     customized form for LAF
>
> -   Sec 2 : If b or e is equal to zero, then L_ab is not continuous. I
>     think these should be limiting cases too.
>
>     If we understand correctly your concern, we can say that if  $ b = 0 $ or $a = 0$,
>     $L_{a,b}$ remains continuous as ${ (\sum  x_i^0})^a = N^a$ and $(
>     \sum x_i^b)^0 = 1$.
>
> -   Sec 4.1 : I fail to understand to what targets input sets are mapped
>     to: one of the feature shown in Fig. 2 or a subset of them.
>     Similarly, I also do not understand what is the global architecture
>     of the approaches. It seems that have 9-dimensional outputs. Please
>     clarify this.
>
>     Here the targets are scalars. We apologize as
>     there was a mistake in the text. The architecture reported in
>     Section 4.1 was the one used for the experiments of Section 4.2. The
>     architecture for experiments in Sec 4.1 is indeed much simpler:
>    $$\begin{aligned}
>     & & \mbox{INPUT} \\{x_1, \ldots, x_N\\}, x_i \in \\{0,1,\ldots, 9\\} \Rightarrow \\\\
>     & & \mbox{Embedding } (10 \times 10) \Rightarrow \mbox{ Sigmoid }
>     \Rightarrow 9 \mbox{ LAF aggregatators } \Rightarrow \\\\
>     & & \mbox{Linear layer } (9 \times 1)\end{aligned}$$
>     We have
>     clarified the architecture in the text and add the full details in
>     the Supplementary Material. Furthermore, we have also expanded
>     Section 2 with a subsection describing explicitly the end-to-end
>     architecture of our approach.
>
> -   Maybe the plots would be more informative if the y-axis was in log
>     scale.
>
>     We used a linear scale to stress the difference between methods can succeed in approximating the target function and
>     those that fail (this difference would be less evident in
>     logscale)
>
> -   Sec 4.2 : Do you use convolutional layers before the aggregation
>     layer ? Are they trained together ?
>
>     The model does not use convolutional layers to process the MNIST digits, but several dense
>     layers (followed by non-linear activations) are used before the
>     actual aggregation. The purpose of those layers is just to obtain a
>     reasonable representation for the MNIST digit pictures. Although one
>     can consider an alternate training, we train the whole architecture
>     end-to-end. The details of the architecture is now in the
>     Supplementary Material. Again we apologize for the mistake in Sec.
>     4.1 and 4.2 about the model architecture.
>
> -   Sec 4.3 : Can you be more specific on the dataset pre-processing ?
>     Why not directly use the raw dataset ?
>
>     The raw ModelNet40 dataset contains CAD model objects. Thus, each object is represented
>     as a quad-mesh, where we have vertices (xyz coordinates) and edges
>     connecting vertices. Considering only the vertices, would end up in
>     having trivial sets for simple structured objects. E.g. TVs,
>     Windows, Tables would consist of a very few points. The strategy
>     here (which was firstly adopted by Zaheer, et.al, 2017), is to first
>     reconstruct the 3D objects from the meshes and then sample uniformly
>     3D points from the object surfaces. Unfortunately, nobody to the
>     best of our knowledge published the preprocessed dataset, and we
>     would like to do it unless some copyright problems
>     arise.

---

### Decision · Program_Chairs · 2021-01-07
**Final Decision**

**Decision:**

Reject

**Comment:**

The authors propose a novel and elegant way for learning parameterized aggregation functions and show that their approach can achieve good performance on several datasets (in many cases outperforming other state-of-the-art methods). This is also appreciated by most of the reviewers. However, there have been several issues regarding the description of the proposed approach and the conducted experiments. These have been partly resolved in the rebuttal phase but should be more carefully assessed in another iteration of reviews.

More specifically: Experiments regarding learning of a single LAF versus multiple LAF should partly be included in the main paper (e.g. Figure 4 showing the performance for different numbers of LAFs). When constructing deep sets in this setting with a similar number of aggregation function it appears not very sensible to me to incorporate the same aggregation function multiple times but one would rather include a set of different fixed aggregation functions (these could be derived from the proposed LAFs). The experiments would also benefit from including set transformers as baselines (set transformers are discussed in the paper but not considered in the experiments as the authors argue that this is an orthogonal approach; while I agree that the goal of set transformers is different, I think there would be big value in understanding how these approaches compare and/or can be combined).

Beyond that I think 	a brief discussion of the related topic of learning pooling operations (e.g., in CNNs) is warranted.

Some reviewers also find that their concerns are only partially addressed in the rebuttal (e.g., regarding the extension from sets to vectors and applications in which the achieved performance differences are bigger).

One point which didn’t come up in the reviews but I would want to see addressed in a future version of the paper is an extended discussion of Figure 4. While there are cases were LAF clearly performs better, there are also cases, where Deep Sets outperform (this seem to be the cases in which the used aggregation units match the considered task). As LAFs can in theory represent these aggregation function it still seems challenging to learn the correct form of the aggregation function — I would appreciate deeper insights an analysis of this aspect. An immediate heuristic solution for many applications for improving performance thus might be to combine LAFs and standard aggregators.

In summary, the submitted paper has big potential but should be carefully revised and the experiments should be extended before the paper is accepted.